# Accurate Image Restoration with Attention Retractable Transformer

**Jiale Zhang**[1], **Yulun Zhang**[2]*, **Jinjin Gu**[3,4], **Yongbing Zhang**[5], **Linghe Kong**[1]*, **Xin Yuan**[6]
[1]Shanghai Jiao Tong University,  [2]ETH Zürich,  [3]Shanghai AI Laboratory,
[4]The University of Sydney,  [5]Harbin Institute of Technology (Shenzhen),  [6]Westlake University

## Abstract

Recently, Transformer-based image restoration networks have achieved promising improvements over convolutional neural networks due to parameter-independent global interactions. To lower computational cost, existing works generally limit self-attention computation within non-overlapping windows. However, each group of tokens are always from a dense area of the image. This is considered as a dense attention strategy since the interactions of tokens are restrained in dense regions. Obviously, this strategy could result in restricted receptive fields. To address this issue, we propose **A**ttention **R**etractable **T**ransformer (ART) for image restoration, which presents both dense and sparse attention modules in the network. The sparse attention module allows tokens from sparse areas to interact and thus provides a wider receptive field. Furthermore, the alternating application of dense and sparse attention modules greatly enhances representation ability of Transformer while providing retractable attention on the input image.We conduct extensive experiments on image super-resolution, denoising, and JPEG compression artifact reduction tasks. Experimental results validate that our proposed ART outperforms state-of-the-art methods on various benchmark datasets both quantitatively and visually. We also provide code and models at https://github.com/gladzhang/ART.

## 1 Introduction

Image restoration aims to recover the high-quality image from its low-quality counterpart and includes a series of computer vision applications, such as image super-resolution (SR) and denoising. It is an ill-posed inverse problem since there are a huge amount of candidates for any original input. Recently, deep convolutional neural networks (CNNs) have been investigated to design various models Kim et al. (2016b); Zhang et al. (2020; 2021b) for image restoration. SRCNN Dong et al. (2014) firstly introduced deep CNN into image SR. Then several representative works utilized residual learning (e.g., EDSR Lim et al. (2017)) and attention mechanism (e.g., RCAN Zhang et al. (2018b)) to train very deep network in image SR. Meanwhile, a number of methods were also proposed for image denoising such as DnCNN Zhang et al. (2017a), RPCNN Xia & Chakrabarti (2020), and BRDNet Tian et al. (2020). These CNN-based networks have achieved remarkable performance.

However, due to parameter-dependent receptive field scaling and content-independent local interactions of convolutions, CNN has limited ability to model long-range dependencies. To overcome this limitation, recent works have begun to introduce self-attention into computer vision systems Hu et al. (2019); Ramachandran et al. (2019); Wang et al. (2020); Zhao et al. (2020). Since Transformer has been shown to achieve state-of-the-art performance in natural language processing Vaswani et al. (2017) and high-level vision tasks Dosovitskiy et al. (2021); Touvron et al. (2021); Wang et al. (2021); Zheng et al. (2021); Chu et al. (2021), researchers have been investigating Transformer-based image restoration networks Yang et al. (2020); Wang et al. (2022b). Chen et al. proposed a pre-trained image processing Transformer named IPT Chen et al. (2021a). Liang et al. proposed a strong baseline model named SwinIR Liang et al. (2021) based on Swin Transformer Liu et al. (2021) for image restoration. Zamir et al. also proposed an efficient Transformer model using U-net structure named Restormer Zamir et al. (2022) and achieved state-of-the-art results on several image restoration tasks. In contrast, higher performance can be achieved when using Transformer.

---

*Corresponding authors: Yulun Zhang, yulun100@gmail.com; Linghe Kong, linghe.kong@sjtu.edu.cn

Despite showing outstanding performance, existing Transformer backbones for image restoration still suffer from serious defects. As we know, SwinIR Liang et al. (2021) takes advantage of shifted window scheme to limit self-attention computation within non-overlapping windows. On the other hand, IPT Chen et al. (2021a) directly splits features into $P \times P$ patches to shrink original feature map $P^2$ times, treating each patch as a token. In short, these methods compute self-attention with shorter token sequences and the tokens in each group are always from a dense area of the image. It is considered as a dense attention strategy, which obviously causes a restricted receptive field. To address this issue, the sparse attention strategy is employed. We extract each group of tokens from a sparse area of the image to provide interactions like previous studies (e.g., GG-Transformer Yu et al. (2021), MaxViT Tu et al. (2022b), Cross-Former Wang et al. (2022a)), but different from them. Our proposed sparse attention module

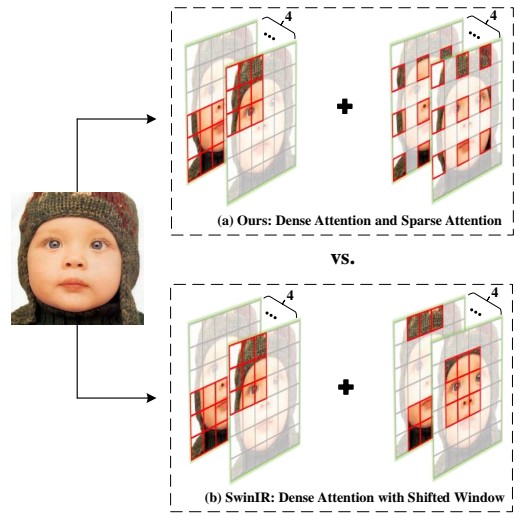

Figure 1: (**a**) Dense attention and sparse attention strategies of our ART. (**b**) Dense attention strategy with shifted window of SwinIR.

focuses on equal-scale features. Besides, We pay more attention to pixel-level information than semantic-level information. Since the sparse attention has not been well proposed to solve the problems in low-level vision fields, our proposed method can bridge this gap.

We further propose Attention Retractable Transformer named ART for image restoration. Following RCAN Zhang et al. (2018b) and SwinIR Liang et al. (2021), we reserve the residual in residual structure Zhang et al. (2018b) for model architecture. Based on joint dense and sparse attention strategies, we design two types of self-attention blocks. We utilize fixed non-overlapping local windows to obtain tokens for the first block named dense attention block (DAB) and sparse grids to obtain tokens for the second block named sparse attention block (SAB). To better understand the difference between our work and SwinIR, we show a visual comparison in Fig. 1. As we can see, the image is divided into four groups and tokens in each group interact with each other. Visibly, the token in our sparse attention block can learn relationships from farther tokens while the one in dense attention block of SwinIR cannot. At the same computational cost, the sparse attention block has stronger ability to compensate for the lack of global information. We consider our dense and sparse attention blocks as successive ones and apply them to extract deep feature. In practice, the alternating application of DAB and SAB can provide retractable attention for the model to capture both local and global receptive field. Our main contributions can be summarized as follows:

- We propose the sparse attention to compensate the defect of mainly using dense attention in existing Transformer-based image restoration networks. The interactions among tokens extracted from a sparse area of an image can bring a wider receptive field to the module.

- We further propose Attention Retractable Transformer (ART) for image restoration. Our ART offers two types of self-attention blocks to obtain retractable attention on the input feature. With the alternating application of dense and sparse attention blocks, the Transformer model can capture local and global receptive field simultaneously.

- We employ ART to train an effective Transformer-based network. We conduct extensive experiments on three image restoration tasks: image super-resolution, denoising, and JPEG compression artifact reduction. Our method achieves state-of-the-art performance.

## 2 RELATED WORK

**Image Restoration.** With the rapid development of CNN, numerous works based on CNN have been proposed to solve image restoration problems Anwar & Barnes (2020); Dudhane et al. (2022); Zamir et al. (2020; 2021); Li et al. (2022); Chen et al. (2021b) and achieved superior performance over conventional restoration approaches Timofte et al. (2013); Michaeli & Irani (2013); He et al. (2010). The pioneering work SRCNN Dong et al. (2014) was firstly proposed for image SR. DnCNN Zhang et al. (2017a) was a representative image denoising method. Following these works, various model

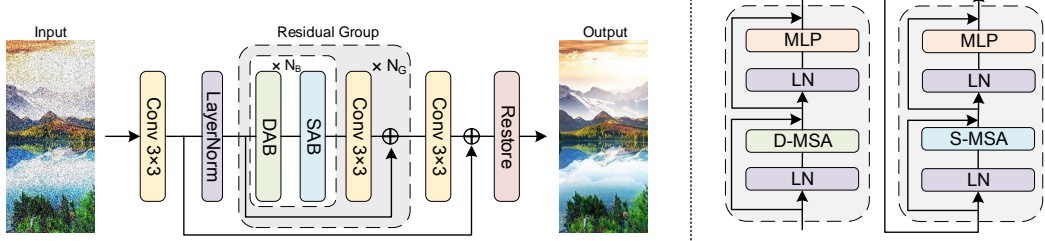

(a) The architecture of ART for image restoration | (b) Two successive attention blocks

Figure 2: (**a**) The architecture of our proposed ART for image restoration. (**b**) The inner structure of two successive attention blocks DAB and SAB with two attention modules D-MSA and S-MSA.

designs and improving techniques have been introduced into the basic CNN frameworks. These techniques include but not limit to the residual structure Kim et al. (2016a); Zhang et al. (2021a), skip connection Zhang et al. (2018b; 2020), dropout Kong et al. (2022), and attention mechanism Dai et al. (2019); Niu et al. (2020). Recently, due to the limited ability of CNN to model long-range dependencies, researchers have started to replace convolution operator with pure self-attention module for image restoration Yang et al. (2020); Liang et al. (2021); Zamir et al. (2022); Chen et al. (2021a).

**Vision Transformer.** Transformer has been achieving impressive performance in machine translation tasks Vaswani et al. (2017). Due to the content-dependent global receptive field, it has been introduced to improve computer vision systems in recent years. Dosovitskiy et al. Dosovitskiy et al. (2021) proposed ViT and introduced Transformer into image recognition by projecting large image patches into token sequences. Tu et al. proposed MaxViT Tu et al. (2022b) as an efficient Vision Transformer while introducing multi-axis attention. Wang et al. proposed CrossFormer Wang et al. (2022a) to build the interactions among long and short distance tokens. Yu et al. proposed GG-Transformer Yu et al. (2021), which performed self-attention on the adaptively-dilated partitions of the input. Inspired by the strong ability to learn long-range dependencies, researches have also investigated the usage of Transformer for low-level vision tasks Yang et al. (2020); Chen et al. (2021a); Liang et al. (2021); Zamir et al. (2022); Wang et al. (2022b). However, existing works still suffer from restricted receptive fields due to mainly using dense attention strategy. Very recently, Tu et al. proposed a MLP-based network named MAXIM Tu et al. (2022a) to introduce dilated spatial communications into image processing. It further demonstrates that the sparse interactions of visual elements are important for solving low-level problems. In our proposed method, we use dense and sparse attention strategies to build network, which can capture wider global interactions. As the sparse attention has not been well proposed to solve the low-level vision problems, our proposed method can bridge this gap.

## 3 PROPOSED METHOD

### 3.1 OVERALL ARCHITECTURE

The overall architecture of our ART is shown in Fig. 2. Following RCAN Zhang et al. (2018b), ART employs residual in residual structure to construct a deep feature extraction module. Given a degraded image $I_{LQ} \in \mathbb{R}^{H \times D \times C_{in}}$ ($H$, $D$, and $C_{in}$ are the height, width, and input channels of the input), ART firstly applies a 3×3 convolutional layer (Conv) to obtain shallow feature $F_0 \in \mathbb{R}^{H \times D \times C}$, where $C$ is the dimension size of the new feature embedding. Next, the shallow feature is normalized and fed into the residual groups, which consist of core Transformer attention blocks. The deep feature is extracted and then passes through another 3×3 Conv to get further feature embeddings $F_1$. Then we use element-wise sum to obtain the final feature map $F_R = F_0 + F_1$. Finally, we employ the restoration module to generate the high-quality image $I_{HQ}$ from the feature map $F_R$.

**Residual Group.** We use $N_G$ successive residual groups to extract the deep feature. Each residual group consists of $N_B$ pairs of attention blocks. We design two successive attention blocks shown in Fig. 2(b). The input feature $x_{l-1}$ passes through layer normalization (LN) and multi-head self-attention (MSA). After adding the shortcut, the output $x'_l$ is fed into the multi-layer perception (MLP). $x_l$ is the final output at the $l$-th block. The process is formulated as

$$x'_l = \text{MSA}(\text{LN}(x_{l-1})) + x_{l-1},$$
$$x_l = \text{MLP}(\text{LN}(x'_l)) + x'_l. \tag{1}$$

Lastly, we also apply a 3×3 convolutional layer to refine the feature embeddings. As shown in Fig 2(a), a residual connection is employed to obtain the final output in each residual group module.

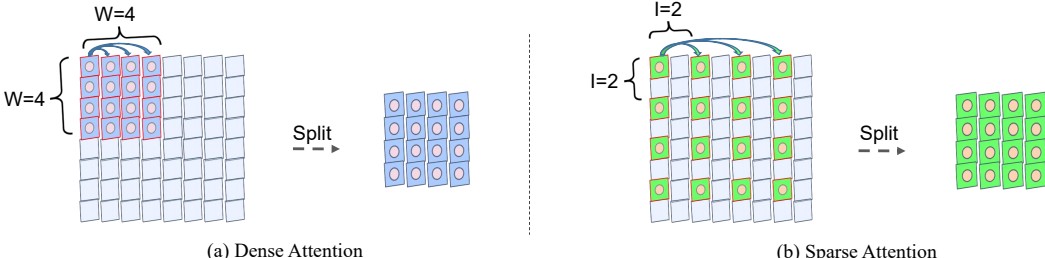

(a) Dense Attention

(b) Sparse Attention

Figure 3: (**a**) Dense attention strategy. Tokens of each group are from a dense area of the image. (**b**) Sparse attention strategy. Tokens of each group are from a sparse area of the image.

**Restoration Module.** The restoration module is applied as the last stage of the framework to obtain the reconstructed image. As we know, image restoration tasks can be divided into two categories according to the usage of upsampling. For image super-resolution, we take advantage of the sub-pixel convolutional layer Shi et al. (2016) to upsample final feature map $F_R$. Next, we use a convolutional layer to get the final reconstructed image $I_{HQ}$. The whole process is formulated as

$$I_{HQ} = \text{Conv}(\text{Upsample}(F_R)). \tag{2}$$

For tasks without upsampling, such as image denoising, we directly use a convolutional layer to reconstruct the high-quality image. Besides, we add the original image to the last output of restoration module for better performance. We formulate the whole process as

$$I_{HQ} = \text{Conv}(F_R) + I_{LQ}. \tag{3}$$

**Loss Function.** We optimize our ART with two types of loss functions. There are various well-studied loss functions, such as $L_2$ loss Dong et al. (2016); Sajjadi et al. (2017); Tai et al. (2017), $L_1$ loss Lai et al. (2017); Zhang et al. (2020), and Charbonnier loss Charbonnier et al. (1994). Same with previous works Zhang et al. (2018b); Liang et al. (2021), we utilize $L_1$ loss for image super-resolution (SR) and Charbonnier loss for image denoising and compression artifact reduction. For image SR, the goal of training ART is to minimize the $L_1$ loss function, which is formulated as

$$\mathcal{L} = \|I_{HQ} - I_G\|_1, \tag{4}$$

where $I_{HQ}$ is the output of ART and $I_G$ is the ground-truth image. For image denoising and JPEG compression artifact reduction, we utilize Charbonnier loss with super-parameter $\varepsilon$ as $10^{-3}$, which is

$$\mathcal{L} = \sqrt{\|I_{HQ} - I_G\|^2 + \varepsilon^2}. \tag{5}$$

## 3.2 ATTENTION RETRACTABLE TRANSFORMER

We elaborate the details about our proposed two types of self-attention blocks in this section. As plotted in Fig. 2(b), the interactions of tokens are concentrated on the multi-head self-attention module (MSA). We formulate the calculation process in MSA as

$$\text{MSA}(X) = \text{Softmax}(\frac{QK^T}{\sqrt{C}})V, \tag{6}$$

where $Q, K, V \in \mathbb{R}^{N \times C}$ are respectively the query, key, and value from the linear projecting of input $X \in \mathbb{R}^{N \times C}$. $N$ is the length of token sequence, and $C$ is the dimension size of each token. Here we assume that the number of heads is 1 to transfer MSA to singe-head self-attention for simplification.

**Multi-head Self Attention.** Given an image with size $H \times D$, vision Transformer firstly splits the raw image into numerous patches. These patches are projected by convolutions with stride size $P$. The new projected feature map $\hat{X} \in \mathbb{R}^{h \times w \times C}$ is prepared with $h = \frac{H}{P}$ and $w = \frac{D}{P}$. Common MSA uses all the tokens extracted from the whole feature map and sends them to self-attention module to learn relationships between each other. It will suffer from high computational cost, which is

$$\Omega(\text{MSA}) = 4hwC^2 + 2(hw)^2 C. \tag{7}$$

To lower the computational cost, existing works generally utilize non-overlapping windows to obtain shorter token sequences. However, they mainly consider the tokens from a dense area of an image. Different from them, we propose the retractable attention strategies, which provide interactions of tokens from not only dense areas but also sparse areas of an image to obtain a wider receptive field.

**Dense Attention.** As shown in Fig. 3(a), dense attention allows each token to interact with a smaller number of tokens from the neighborhood position of a non-overlapping $W \times W$ window. All tokens

| Methods | Solving problems | Structure | Interval of extracted tokens | Representation of tokens | Using long-distance residual connection |
|---|---|---|---|---|---|
| GG-Transformer Yu et al. (2021) | High-level | Pyramid | Changed | Semantic-level | No |
| MaxViT Tu et al. (2022b) | High-level | Pyramid | Changed | Semantic-level | No |
| CrossFormer Wang et al. (2022a) | High-level | Pyramid | Changed | Semantic-level | No |
| ART (Ours) | Low-level | Isotropic | Unchanged | Pixel-level | Yes |

Table 1: Comparison to related works. The differences between our ART with other works.

are split into several groups and each group has $W \times W$ tokens. We apply these groups to compute self-attention for $\frac{h}{W} \times \frac{w}{W}$ times and the computational cost of new module named D-MSA is

$$\Omega(\text{D-MSA}) = (4W^2C^2 + 2W^4C) \times \frac{h}{W} \times \frac{w}{W} = 4hwC^2 + 2W^2hwC. \tag{8}$$

**Sparse Attention.** Meanwhile, as shown in Fig. 3(b), we propose sparse attention to allow each token to interact with a smaller number of tokens, which are from sparse positions with interval size $I$. After that, the updates of all tokens are also split into several groups and each group has $\frac{h}{I} \times \frac{w}{I}$ tokens. We further utilize these groups to compute self-attention for $I \times I$ times. We name the new multi-head self-attention module as S-MSA and the corresponding computational cost is

$$\Omega(\text{S-MSA}) = (4\frac{h}{I} \times \frac{w}{I}C^2 + 2(\frac{h}{I} \times \frac{w}{I})^2C) \times I \times I = 4hwC^2 + 2\frac{h}{I}\frac{w}{I}hwC. \tag{9}$$

By contrast, our proposed D-MSA and S-MSA modules have lower computational cost since $W^2 \ll hw$ and $\frac{h}{I}\frac{w}{I} < hw$. After computing all groups, the outputs are further merged to form original-size feature map. In practice, we apply these two attention strategies to design two types of self-attention blocks named as dense attention block (DAB) and sparse attention block (SAB) as plotted in Fig. 2.

**Successive Attention Blocks.** We propose the alternating application of these two blocks. As the local interactions have higher priority, we fix the order of DAB in front of SAB. Besides, we provide the long-distance residual connection between each three pairs of blocks. We show the effectiveness of this joint application with residual connection in the supplementary material.

**Attention Retractable Transformer.** We demonstrate that the application of these two blocks enables our model to capture local and global receptive field simultaneously. We treat the successive attention blocks as a whole and get a new type of Transformer named Attention Retractable Transformer, which can provide interactions for both local dense tokens and global sparse tokens.

### 3.3 DIFFERENCES TO RELATED WORKS

We summarize the differences between our proposed approach, ART with the closely related works in Tab. 1. We conclude them as three points. **(1) Different tasks.** GG-Transformer Yu et al. (2021), MaxViT Tu et al. (2022b) and CrossFormer Wang et al. (2022a) are proposed to solve high-level vision problems. Our ART is the only one to employ the sparse attention in low-level vision fields. **(2) Different designs of sparse attention.** In the part of attention, GG-Transformer utilizes the adaptively-dilated partitions, MaxViT utilizes the fixed-size grid attention and CrossFormer utilizes the cross-scale long-distance attention. As the layers get deeper, the interval of tokens from sparse attention becomes smaller and the channels of tokens become larger. Therefore, each token learns more semantic-level information. In contrast, the interval and the channel dimension of tokens in our ART keep unchanged and each token represents the accurate pixel-level information. **(3) Different model structures.** Different from these works using Pyramid model structure, our proposed ART enjoys an Isotropic structure. Besides, we provide the long-distance residual connection between several Transformer encoders, which enables the feature of deep layers to reserve more low-frequency information from shallow layers. More discussion can be found in the supplementary material.

### 3.4 IMPLEMENTATION DETAILS

Some details about how to apply our ART to construct image restoration model are introduced here. Firstly, the residual group number, DAB number, and SAB number in each group are set as 6, 3, and 3. Secondly, all the convolutional layers are equipped with $3 \times 3$ kernel, 1-length stride, and 1-length padding, so the height and width of feature map remain unchanged. In practice, we treat $1 \times 1$ patch as a token. Besides, we set the channel dimension as 180 for most layers except for the shallow feature extraction and the image reconstruction process. Thirdly, the window size in DAB is set as 8 and the interval size in SAB is adjustable according to different tasks, which is discussed in Sec. 4.2. Lastly, to adjust the division of windows and sparse grids, we use padding and mask strategies to the input feature map of self-attention, so that the number of division is always an integer.

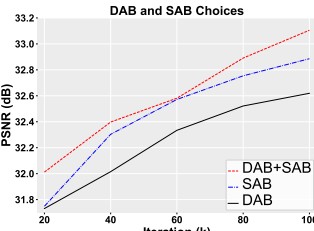 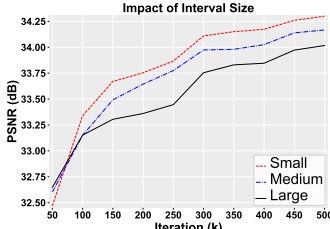 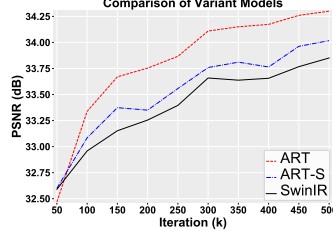

Figure 4: **Left:** PSNR (dB) comparison of our ART using all dense attention block (DAB), using all sparse attention block (SAB), and using alternating DAB and SAB. **Middle:** PSNR (dB) comparison of our ART using large interval size in sparse attention block which is $(8, 8, 8, 8, 8, 8)$ for six residual groups, using medium interval size which is $(8, 8, 6, 6, 4, 4)$, and using small interval size which is $(4, 4, 4, 4, 4, 4)$. **Right:** PSNR (dB) comparison of SwinIR, ART-S, and ART.

## 4 EXPERIMENTAL RESULTS

### 4.1 EXPERIMENTAL SETTINGS

**Data and Evaluation.** We conduct experiments on three image restoration tasks, including image SR, denoising, and JPEG Compression Artifact Reduction (CAR). For image SR, following previous works Zhang et al. (2018b); Haris et al. (2018), we use DIV2K Timofte et al. (2017) and Flickr2K Lim et al. (2017) as training data, Set5 Bevilacqua et al. (2012), Set14 Zeyde et al. (2010), B100 Martin et al. (2001), Urban100 Huang et al. (2015), and Manga109 Matsui et al. (2017) as test data. For image denoising and JPEG CAR, same as SwinIR Liang et al. (2021), we use training data: DIV2K, Flickr2K, BSD500 Arbelaez et al. (2010), and WED Ma et al. (2016). We use BSD68 Martin et al. (2001), Kodak24 Franzen (1999), McMaster Zhang et al. (2011), and Urban100 as test data of image denoising. Classic5 Foi et al. (2007) and LIVE1 Sheikh et al. (2006) are test data of JPEG CAR. Note that we crop large-size input image into $200\times200$ partitions with overlapping pixels during inference. Following Lim et al. (2017), we adopt the self-ensemble strategy to further improve the performance of our ART and name it as ART+. We evaluate experimental results with PSNR and SSIM Wang et al. (2004) values on Y channel of images transformed to YCbCr space.

**Training Settings.** Data augmentation is performed on the training data through horizontal flip and random rotation of $90°$, $180°$, and $270°$. Besides, we crop the original images into $64\times64$ patches as the basic training inputs for image SR, $128\times128$ patches for image denoising, and $126\times126$ patches for JPEG CAR. We resize the training batch to 32 for image SR, and 8 for image denoising and JPEG CAR in order to make a fair comparison. We choose ADAM Kingma & Ba (2015) to optimize our ART model with $\beta_1 = 0.9$, $\beta_2 = 0.999$, and zero weight decay. The initial learning rate is set as $2\times10^{-4}$ and is reduced by half as the training iteration reaches a certain number. Taking image SR as an example, we train ART for total 500k iterations and adjust learning rate to half when training iterations reach 250k, 400k, 450k, and 475k, where 1k means one thousand. Our ART is implemented on PyTorch Paszke et al. (2017) with 4 NVIDIA RTX8000 GPUs.

### 4.2 ABLATION STUDY

For ablation experiments, we train our models for image super-resolution ($\times2$) based on DIV2K and Flicke2K datasets. The results are evaluated on Urban100 benchmark dataset.

**Design Choices for DAB and SAB.** We demonstrate the necessity for simultaneous usage of dense attention block (DAB) and sparse attention block (SAB) by conducting ablation study. We set three different experiment conditions, which are using 6 DABs, 6 SABs, and 3 pairs of alternating DAB and SAB. We keep the rest of experiment environment the same and train all models within 100k iterations. The experimental results are shown in Fig. 4(Left). As we can see, only using DAB or SAB suffers from poor performance, because they lack either global receptive field or local receptive field. On the other hand, the structure of SAB following DAB brings higher performance. It validates that both local contextual interactions and global sparse interactions are important for improving strong representation ability of Transformer by obtaining retractable attention on the input feature.

**Impact of Interval Size.** The interval size in sparse attention block has a vital impact on the performance of our ART. In fact, if the interval size is set as 1, it will be transferred to full attention. Generally, a smaller interval means wider receptive fields but higher computational cost. We compare the experimental results under different interval settings in Fig. 4(Middle). As we can see, smaller

| Method | Scale | Set5 PSNR | Set5 SSIM | Set14 PSNR | Set14 SSIM | B100 PSNR | B100 SSIM | Urban100 PSNR | Urban100 SSIM | Manga109 PSNR | Manga109 SSIM |
|---|---|---|---|---|---|---|---|---|---|---|---|
| EDSR Lim et al. (2017) | ×2 | 38.11 | 0.9602 | 33.92 | 0.9195 | 32.32 | 0.9013 | 32.93 | 0.9351 | 39.10 | 0.9773 |
| RCAN Zhang et al. (2018b) | ×2 | 38.27 | 0.9614 | 34.12 | 0.9216 | 32.41 | 0.9027 | 33.34 | 0.9384 | 39.44 | 0.9786 |
| SAN Dai et al. (2019) | ×2 | 38.31 | 0.9620 | 34.07 | 0.9213 | 32.42 | 0.9028 | 33.10 | 0.9370 | 39.32 | 0.9792 |
| SRFBN Li et al. (2019) | ×2 | 38.11 | 0.9609 | 33.82 | 0.9196 | 32.29 | 0.9010 | 32.62 | 0.9328 | 39.08 | 0.9779 |
| HAN Niu et al. (2020) | ×2 | 38.27 | 0.9614 | 34.16 | 0.9217 | 32.41 | 0.9027 | 33.35 | 0.9385 | 39.46 | 0.9785 |
| IGNN Zhou et al. (2020) | ×2 | 38.24 | 0.9613 | 34.07 | 0.9217 | 32.41 | 0.9025 | 33.23 | 0.9383 | 39.35 | 0.9786 |
| CSNLN Mei et al. (2020) | ×2 | 38.28 | 0.9616 | 34.12 | 0.9223 | 32.40 | 0.9024 | 33.25 | 0.9386 | 39.37 | 0.9785 |
| RFANet Liu et al. (2020) | ×2 | 38.26 | 0.9615 | 34.16 | 0.9220 | 32.41 | 0.9026 | 33.33 | 0.9389 | 39.44 | 0.9783 |
| NLSA Mei et al. (2021) | ×2 | 38.34 | 0.9618 | 34.08 | 0.9231 | 32.43 | 0.9027 | 33.42 | 0.9394 | 39.59 | 0.9789 |
| IPT Chen et al. (2021a) | ×2 | 38.37 | N/A | 34.43 | N/A | 32.48 | N/A | 33.76 | N/A | N/A | N/A |
| SwinIR Liang et al. (2021) | ×2 | 38.42 | 0.9623 | 34.46 | 0.9250 | 32.53 | 0.9041 | 33.81 | 0.9427 | 39.92 | 0.9797 |
| **ART-S (ours)** | ×2 | 38.48 | 0.9625 | 34.50 | 0.9258 | 32.53 | 0.9043 | 34.02 | 0.9437 | 40.11 | 0.9804 |
| **ART (ours)** | ×2 | 38.56 | 0.9629 | 34.59 | 0.9267 | 32.58 | 0.9048 | 34.30 | 0.9452 | 40.24 | 0.9808 |
| **ART+ (ours)** | ×2 | 38.59 | 0.9630 | 34.68 | 0.9269 | 32.60 | 0.9050 | 34.41 | 0.9457 | 40.33 | 0.9810 |
| EDSR Lim et al. (2017) | ×3 | 34.65 | 0.9280 | 30.52 | 0.8462 | 29.25 | 0.8093 | 28.80 | 0.8653 | 34.17 | 0.9476 |
| RCAN Zhang et al. (2018b) | ×3 | 34.74 | 0.9299 | 30.65 | 0.8482 | 29.32 | 0.8111 | 29.09 | 0.8702 | 34.44 | 0.9499 |
| SAN Dai et al. (2019) | ×3 | 34.75 | 0.9300 | 30.59 | 0.8476 | 29.33 | 0.8112 | 28.93 | 0.8671 | 34.30 | 0.9494 |
| SRFBN Li et al. (2019) | ×3 | 34.70 | 0.9292 | 30.51 | 0.8461 | 29.24 | 0.8084 | 28.73 | 0.8641 | 34.18 | 0.9481 |
| HAN Niu et al. (2020) | ×3 | 34.75 | 0.9299 | 30.67 | 0.8483 | 29.32 | 0.8110 | 29.10 | 0.8705 | 34.48 | 0.9500 |
| IGNN Zhou et al. (2020) | ×3 | 34.72 | 0.9298 | 30.66 | 0.8484 | 29.31 | 0.8105 | 29.03 | 0.8696 | 34.39 | 0.9496 |
| CSNLN Mei et al. (2020) | ×3 | 34.74 | 0.9300 | 30.66 | 0.8482 | 29.33 | 0.8105 | 29.13 | 0.8712 | 34.45 | 0.9502 |
| RFANet Liu et al. (2020) | ×3 | 34.79 | 0.9300 | 30.67 | 0.8487 | 29.34 | 0.8115 | 29.15 | 0.8720 | 34.59 | 0.9506 |
| NLSA Mei et al. (2021) | ×3 | 34.85 | 0.9306 | 30.70 | 0.8485 | 29.34 | 0.8117 | 29.25 | 0.8726 | 34.57 | 0.9508 |
| IPT Chen et al. (2021a) | ×3 | 34.81 | N/A | 30.85 | N/A | 29.38 | N/A | 29.49 | N/A | N/A | N/A |
| SwinIR Liang et al. (2021) | ×3 | 34.97 | 0.9318 | 30.93 | 0.8534 | 29.46 | 0.8145 | 29.75 | 0.8826 | 35.12 | 0.9537 |
| **ART-S (ours)** | ×3 | 34.98 | 0.9318 | 30.94 | 0.8530 | 29.45 | 0.8146 | 29.86 | 0.8830 | 35.22 | 0.9539 |
| **ART (ours)** | ×3 | 35.07 | 0.9325 | 31.02 | 0.8541 | 29.51 | 0.8159 | 30.10 | 0.8871 | 35.39 | 0.9548 |
| **ART+ (ours)** | ×3 | 35.11 | 0.9327 | 31.05 | 0.8545 | 29.53 | 0.8162 | 30.22 | 0.8883 | 35.51 | 0.9552 |
| EDSR Lim et al. (2017) | ×4 | 32.46 | 0.8968 | 28.80 | 0.7876 | 27.71 | 0.7420 | 26.64 | 0.8033 | 31.02 | 0.9148 |
| RCAN Zhang et al. (2018b) | ×4 | 32.63 | 0.9002 | 28.87 | 0.7889 | 27.77 | 0.7436 | 26.82 | 0.8087 | 31.22 | 0.9173 |
| SAN Dai et al. (2019) | ×4 | 32.64 | 0.9003 | 28.92 | 0.7888 | 27.78 | 0.7436 | 26.79 | 0.8068 | 31.18 | 0.9169 |
| SRFBN Li et al. (2019) | ×4 | 32.47 | 0.8983 | 28.81 | 0.7868 | 27.72 | 0.7409 | 26.60 | 0.8015 | 31.15 | 0.9160 |
| HAN Niu et al. (2020) | ×4 | 32.64 | 0.9002 | 28.90 | 0.7890 | 27.80 | 0.7442 | 26.85 | 0.8094 | 31.42 | 0.9177 |
| IGNN Zhou et al. (2020) | ×4 | 32.57 | 0.8998 | 28.85 | 0.7891 | 27.77 | 0.7434 | 26.84 | 0.8090 | 31.28 | 0.9182 |
| CSNLN Mei et al. (2020) | ×4 | 32.68 | 0.9004 | 28.95 | 0.7888 | 27.80 | 0.7439 | 27.22 | 0.8168 | 31.43 | 0.9201 |
| RFANet Liu et al. (2020) | ×4 | 32.66 | 0.9004 | 28.88 | 0.7894 | 27.79 | 0.7442 | 26.92 | 0.8112 | 31.41 | 0.9187 |
| NLSA Mei et al. (2021) | ×4 | 32.59 | 0.9000 | 28.87 | 0.7891 | 27.78 | 0.7444 | 26.96 | 0.8109 | 31.27 | 0.9184 |
| IPT Chen et al. (2021a) | ×4 | 32.64 | N/A | 29.01 | N/A | 27.82 | N/A | 27.26 | N/A | N/A | N/A |
| SwinIR Liang et al. (2021) | ×4 | 32.92 | 0.9044 | 29.09 | 0.7950 | 27.92 | 0.7489 | 27.45 | 0.8254 | 32.03 | 0.9260 |
| **ART-S (ours)** | ×4 | 32.86 | 0.9029 | 29.09 | 0.7942 | 27.91 | 0.7489 | 27.54 | 0.8261 | 32.13 | 0.9263 |
| **ART (ours)** | ×4 | 33.04 | 0.9051 | 29.16 | 0.7958 | 27.97 | 0.7510 | 27.77 | 0.8321 | 32.31 | 0.9283 |
| **ART+ (ours)** | ×4 | 33.07 | 0.9055 | 29.20 | 0.7964 | 27.99 | 0.7513 | 27.89 | 0.8339 | 32.45 | 0.9291 |

Table 2: PSNR (dB)/SSIM comparisons for image super-resolution on five benchmark datasets. We color best and second best results in red and blue.

| Method | EDSR | RCAN | SRFBN | HAN | CSNLN | SwiIR | ART-S (ours) | ART (ours) |
|---|---|---|---|---|---|---|---|---|
| Params (M) | 43.09 | 15.59 | 3.63 | 16.07 | 7.16 | 11.90 | 11.87 | 16.55 |
| Mult-Adds (G) | 1,286 | 407 | 498 | 420 | 103,640 | 336 | 392 | 782 |
| PSNR on Urban100 (dB) | 26.64 | 26.82 | 26.60 | 26.85 | 27.22 | 27.45 | 27.54 | 27.77 |
| PSNR on Manga109 (dB) | 31.02 | 31.22 | 31.15 | 31.42 | 31.43 | 32.03 | 32.13 | 32.31 |

Table 3: Model size comparisons (×4 SR). Output size is 3×640×640 for Mult-Adds calculation.

intervals bring more performance gains. To keep the balance between accuracy and complexity, we set the interval size of 6 residual groups as $(4, 4, 4, 4, 4, 4)$ for image SR, $(16, 16, 12, 12, 8, 8)$ for image denoising, and $(18, 18, 13, 13, 7, 7)$ for JPEG CAR in the following comparative experiments.

**Comparison of Variant Models.** We provide a new version of our model for fair comparisons and name it ART-S. Different from ART, the MLP ratio in ART-S is set to 2 (4 in ART) and the interval size is set to 8. We demonstrate that ART-S has comparable model size with SwinIR. We provide the PSNR comparison results in Fig. 4(Right). As we can see, our ART-S achieves better performance than SwinIR. More comparative results can be found in following experiment parts.

## 4.3 IMAGE SUPER-RESOLUTION

We provide comparisons of our proposed ART with representative image SR methods, including CNN-based networks: EDSR Lim et al. (2017), RCAN Zhang et al. (2018b), SAN Dai et al. (2019), SRFBN Li et al. (2019), HAN Niu et al. (2020), IGNN Zhou et al. (2020), CSNLN Mei et al. (2020), RFANet Liu et al. (2020), NLSA Mei et al. (2021), and Transformer-based networks: IPT Chen et al. (2021a) and SwinIR Liang et al. (2021). Note that IPT is a pre-trained model, which is trained on ImageNet benchmark dataset. All the results are provided by publicly available code and data. Quantitative and visual comparisons are provided in Tab. 2 and Fig. 5.

**Quantitative Comparisons.** We present PSNR/SSIM comparison results for ×2, ×3, and ×4 image SR in Tab. 2. As we can see, our ART achieves the best PSNR/SSIM performance on all five benchmark datasets. Using self-ensemble, ART+ gains even better results. Compared with existing state-of-the-art method SwinIR, our ART obtains better gains across all scale factors, indicating that

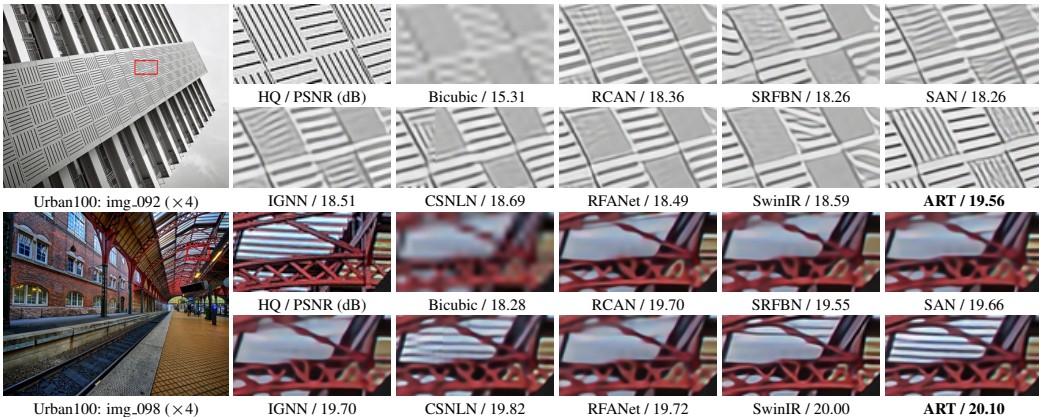

Figure 5: Visual comparison with challenging examples on image super-resolution (×4).

our proposed joint dense and sparse attention blocks enable Transformer stronger representation ability. Despite showing better performance than CNN-based networks, another Transformer-based network IPT is not as good as ours. It is validated that our proposed ART becomes a new promising Transformer-based network for image SR.

**Retractable vs. Dense Attention.** We further show a typical visual comparison with SwinIR in Fig. 6. As SwinIR mainly utilizes dense attention strategy, it restores wrong texture structures under the influence of close patches with mainly vertical lines. However, our ART can reconstruct the right texture, thanks to the wider receptive field provided by sparse attention strategy. Visibly, the patch is able to interact with farther patches

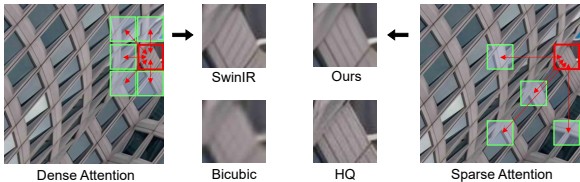

Figure 6: Visual comparison (×4) of SwinIR and Ours.

with similar horizontal lines so that it can be reconstructed clearly. This comparison demonstrates the advantage of retractable attention and its strong ability to restore high-quality outputs.

**Model Size Comparisons.** Table 3 provides comparisons of parameters number and Mult-Adds of different networks, which include existing state-of-the-art methods. We calculate the Mult-Adds assuming that the output size is 3×640×640 under ×4 image SR. Compared with previous CNN-based networks, our ART has comparable parameter number and Mult-Adds but achieves high performance. Besides, we can see that our ART-S has less parameters and Mult-Adds than most of the compared methods. The model size of ART-S is similar with SwinIR. However, ART-S still achieves better performance gains than all compared methods except our ART. It indicates that our method is able to achieve promising performance at an acceptable computational and memory cost.

**Visual Comparisons.** We also provide some challenging examples for visual comparison (×4) in Fig. 5. We can see that our ART is able to alleviate heavy blurring artifacts while restoring detailed edges and textures. Compared with other methods, ART obtains visually pleasing results by recovering more high-frequency details. It indicates that ART preforms better for image SR.

## 4.4 IMAGE DENOISING

We show color image denoising results to compare our ART with representative methods in Tab. 4. These methods are CBM3D Dabov et al. (2007), IRCNN Zhang et al. (2017b), FFDNet Zhang et al. (2018a), DnCNN Zhang et al. (2017a), RNAN Zhang et al. (2019), RDN Zhang et al. (2020), IPT Chen et al. (2021a), DRUNet Zhang et al. (2021a), P3AN Hu et al. (2021), SwinIR Liang et al. (2021), and Restormer Zamir et al. (2022). Following most recent works, we set the noise level to 15, 25, and 50. We also shows visual comparisons of challenging examples in Fig. 7.

**Quantitative Comparisons.** Table 4 shows PSNR results of color image denoising. As we can see, our ART achieves the highest performance across all compared methods on three datasets except Kodak24. Even better results are obtained by ART+ using self-ensemble. Particularly, it obtains better gains than the state-of-the-art model Restormer Zamir et al. (2022) by up to 0.25dB on Urban100. Restormer also has restricted receptive fields and thus has difficulty in some challenging cases. In conclusion, these comparisons indicate that our ART also has strong ability in image denoising.

| Method | BSD68 | | | Kodak24 | | | McMaster | | | Urban100 | | |
|---|---|---|---|---|---|---|---|---|---|---|---|---|
| | $\sigma$=15 | $\sigma$=25 | $\sigma$=50 | $\sigma$=15 | $\sigma$=25 | $\sigma$=50 | $\sigma$=15 | $\sigma$=25 | $\sigma$=50 | $\sigma$=15 | $\sigma$=25 | $\sigma$=50 |
| CBM3D Dabov et al. (2007) | N/A | N/A | 27.38 | N/A | N/A | 28.63 | N/A | N/A | N/A | N/A | N/A | 27.94 |
| IRCNN Zhang et al. (2017b) | 33.86 | 31.16 | 27.86 | 34.69 | 32.18 | 28.93 | 34.58 | 32.18 | 28.91 | 33.78 | 31.20 | 27.70 |
| FFDNet Zhang et al. (2018a) | 33.87 | 31.21 | 27.96 | 34.63 | 32.13 | 28.98 | 34.66 | 32.35 | 29.18 | 33.83 | 31.40 | 28.05 |
| DnCNN Zhang et al. (2017a) | 33.90 | 31.24 | 27.95 | 34.60 | 32.14 | 28.95 | 33.45 | 31.52 | 28.62 | 32.98 | 30.81 | 27.59 |
| RNAN Zhang et al. (2019) | N/A | N/A | 28.27 | N/A | N/A | 29.58 | N/A | N/A | 29.72 | N/A | N/A | 29.08 |
| RDN Zhang et al. (2020) | N/A | N/A | 28.31 | N/A | N/A | 29.66 | N/A | N/A | N/A | N/A | N/A | 29.38 |
| IPT Chen et al. (2021a) | N/A | N/A | 28.39 | N/A | N/A | 29.64 | N/A | N/A | 29.98 | N/A | N/A | 29.71 |
| DRUNet Zhang et al. (2021a) | 34.30 | 31.69 | 28.51 | 35.31 | 32.89 | 29.86 | 35.40 | 33.14 | 30.08 | 34.81 | 32.60 | 29.61 |
| P3AN Hu et al. (2021) | N/A | N/A | 28.37 | N/A | N/A | N/A | N/A | N/A | N/A | N/A | N/A | 29.51 |
| SwinIR Liang et al. (2021) | 34.42 | 31.78 | 28.56 | 35.34 | 32.89 | 29.79 | 35.61 | 33.20 | 30.22 | 35.13 | 32.90 | 29.82 |
| Restormer Zamir et al. (2022) | 34.40 | 31.79 | 28.60 | 35.47 | 33.04 | 30.01 | 35.61 | 33.34 | 30.30 | 35.13 | 32.96 | 30.02 |
| **ART (ours)** | 34.46 | 31.84 | 28.63 | 35.39 | 32.95 | 29.87 | 35.68 | 33.41 | 30.31 | 35.29 | 33.14 | 30.19 |
| **ART+ (ours)** | 34.47 | 31.85 | 28.65 | 35.41 | 32.98 | 29.89 | 35.71 | 33.44 | 30.35 | 35.34 | 33.20 | 30.27 |

Table 4: PSNR (dB) comparisons. The best and second best results are in red and blue.

| Dataset | q | RNAN | | RDN | | DRUNet | | SwinIR | | ART (ours) | | ART+ (ours) | |
|---|---|---|---|---|---|---|---|---|---|---|---|---|---|
| | | PSNR | SSIM | PSNR | SSIM | PSNR | SSIM | PSNR | SSIM | PSNR | SSIM | PSNR | SSIM |
| Classic5 | 10 | 29.96 | 0.8178 | 30.00 | 0.8188 | 30.16 | 0.8234 | 30.27 | 0.8249 | 30.27 | 0.8258 | 30.32 | 0.8263 |
| | 30 | 33.38 | 0.8924 | 33.43 | 0.8930 | 33.59 | 0.8949 | 33.73 | 0.8961 | 33.74 | 0.8964 | 33.78 | 0.8967 |
| | 40 | 34.27 | 0.9061 | 34.27 | 0.9061 | 34.41 | 0.9075 | 34.52 | 0.9082 | 34.55 | 0.9086 | 34.58 | 0.9089 |
| LIVE1 | 10 | 29.63 | 0.8239 | 29.67 | 0.8247 | 29.79 | 0.8278 | 29.86 | 0.8287 | 29.89 | 0.8300 | 29.92 | 0.8305 |
| | 30 | 33.45 | 0.9149 | 33.51 | 0.9153 | 33.59 | 0.9166 | 33.69 | 0.9174 | 33.71 | 0.9178 | 33.74 | 0.9181 |
| | 40 | 34.47 | 0.9299 | 34.51 | 0.9302 | 34.58 | 0.9312 | 34.67 | 0.9317 | 34.70 | 0.9322 | 34.73 | 0.9324 |

Table 5: PSNR (dB)/SSIM comparisons. The best and second best results are in red and blue.

**Visual Comparisons.** The visual comparison for color image denoising of different methods is shown in Fig. 7. Our ART can preserve detailed textures and high-frequency components and remove heavy noise corruption. Compared with other methods, it has better performance to restore clean and crisp images. It demonstrates that our ART is also suitable for image denoising.

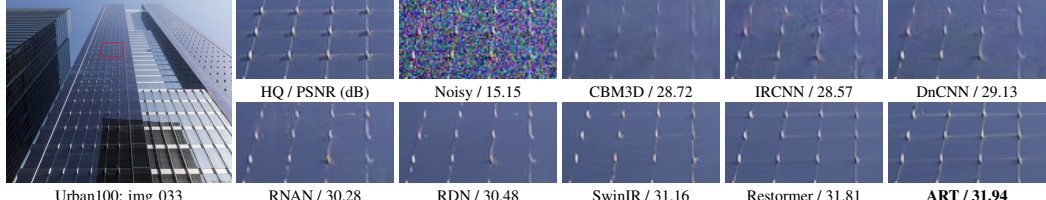

Urban100: img_033    RNAN / 30.28    RDN / 30.48    SwinIR / 31.16    Restormer / 31.81    **ART / 31.94**

Figure 7: Visual comparison with challenging examples on color image denoising ($\sigma$=50).

### 4.5 JPEG COMPRESSION ARTIFACT REDUCTION

We compare our ART with state-of-the-art JPEG CAR methods: RNAN Zhang et al. (2019), RDN Zhang et al. (2020), DRUNet Zhang et al. (2021a), and SwinIR Liang et al. (2021). Following most recent works, we set the compression quality factors of original images to 40, 30, and 10. We provide the PSNR and SSIM comparison results in Table 5.

**Quantitative Comparisons.** Table 5 shows the PSNR/SSIM comparisons of our ART with existing state-of-the-art methods. We can see that our proposed method has the best performance. Better results are achieved by ART+ using self-ensemble. These results indicate that our ART also performs outstandingly when solving image compression artifact reduction problems.

## 5 CONCLUSION

In this work, we propose Attention Retractable Transformer for image restoration named ART, which offers two types of self-attention blocks to enhance the Transformer representation ability. Most previous image restoration Transformer backbones mainly utilize dense attention modules to alleviate self-attention computation within non-overlapping regions and thus suffer from restricted receptive fields. Without introducing additional computational cost, we employ the sparse attention mechanism to enable tokens from sparse areas of the image to interact with each other. In practice, the alternating application of dense and sparse attention modules is able to provide retractable attention for the model and bring promising improvement. Experiments on image SR, denoising, and JPEG CAR tasks validate that our method achieves state-of-the-art results on various benchmark datasets both quantitatively and visually. In future work, we will try to apply our proposed method to more image restoration tasks, like image deraining, deblurring, dehazing, and so on. We will further explore the potential of sparse attention in solving low-level vision problems.

## ACKNOWLEDGMENTS

This work was supported in part by NSFC grant 62141220, 61972253, U1908212, 62172276, 61972254, the Program for Professor of Special Appointment (Eastern Scholar) at Shanghai Institutions of Higher Learning, the National Natural Science Foundation of China under Grant No. 62271414, Zhejiang Provincial Natural Science Foundation of China under Grant No. LR23F010001. This work was also supported by the Shenzhen Science and Technology Project (JCYJ20200109142808034), and in part by Guangdong Special Support (2019TX05X187). Xin Yuan would like to thank Research Center for Industries of the Future (RCIF) at Westlake University for supporting this work.

## REPRODUCIBILITY STATEMENT

We provide the reproducibility statement of our proposed method in this section. We introduce the model architecture and core dense and sparse attention modules in Sec. 3. Besides, we also give the implementation details. In Sec. 4.1, we provide the detailed experiment settings. To ensure the reproducibility, we provide the source code and pre-trained models at the website[1]. Everyone can run our code to check the training and testing process according to the given instructions. At the website, the pre-trained models are provided to verify the validity of corresponding results. More details please refer to the website or the submitted supplementary materials.

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
