# OpenReview forum: "Accurate Image Restoration with Attention Retractable Transformer"
_ICLR.cc/2023/Conference — ICLR 2023 notable top 25%_

### Official Review · Reviewer_f4x2 · 2022-10-23

**Confidence:** 5
**Correctness:** 4
**Technical Novelty And Significance:** 3
**Empirical Novelty And Significance:** 3
**Recommendation:** 8

**Clarity, Quality, Novelty And Reproducibility:**

The paper is high-quality, clear, and novel. Furthermore, the authors provide source code and pretrained models, which make it be easily reproducible.

**Strength And Weaknesses:**

Strength:

1. The paper is well-written and easy to follow. It is obvious that the authors pay much attention to most details.

2. The idea about attention retractable Transformer (ART) is motivated by some observations from image restoration tasks and is more specifically designed. The authors provide illustrations of differences between the proposed method and others, like in Figure 1 and Table 1.

3. They propose a sparse attention to compensate the defect of dense attention used in most Transformer-based image restoration methods. In the sparse area of an image, the token interactions help enlarge the receptive field of the module.

4. Alternating sparse and dense self-attention blocks helps provide retractable attention. Then, the ART model can capture both local and global information simultaneously.

5. The ablation study experiments well support the claimed contributions. The main results with promising performance show the superior and effectiveness of the proposed method.

6. In the supplementary file, the authors not only provide more results (e.g., real image denoising), but also provide the source code, pretained models, and detailed readme information (e.g., scripts). Those supplements make the paper stronger and more solid.

Weakness:

1. For image denoising, the authors mainly provide color image denoising results in the main paper and real image denoising in supplementary file. It would be better to include real image denoising in the paper too.

2. In the demo code, the authors mainly provide testing scripts. Will it be possible to give all the training details in the future? That would be good to the image restoration community.

3. Some details are not very clear. For image denoising, did the authors use the framework and/or training strategy (e.g., progressive training) as used in Restormer?

**Summary Of The Paper:**

The authors proposed an attention retractable Transformer (ART) for accurate image restoration, which consists of dense and sparse attention modules. The experiments are extensive and include several general image restoration tasks, where the proposed method achieves SOTA performance.

**Summary Of The Review:**

A well-prepared paper with solid and extensive experiments.

---

> ### Author Response · Authors · 2022-11-14
> **Response to Reviewer f4x2 (denoted as R4)**
>
> `Q1`: For image denoising, the authors mainly provide color image denoising results in the main paper and real image denoising in supplementary file. It would be better to include real image denoising in the paper too.
>
> `A1`: Thanks for your suggestions. **We also think that it is better to adjust the position** of the introduction about real image denoising task to the main paper. Due to the limited space, we will do this in a future revised version.
>
>
> `Q2`: In the demo code, the authors mainly provide testing scripts. Will it be possible to give all the training details in the future? That would be good to the image restoration community.
>
> `A2`: Thanks for your suggestions. We agree that the open-source code is important and helpful for pushing forward the development of this field. **We promise to give all the training and testing details in the future.** To ensure the reproducibility, we will provide the complete training and testing instructions.
>
>
> `Q3`: Some details are not very clear. For image denoising, did the authors use the framework and/or training strategy (e.g., progressive training) as used in Restormer?
>
> `A3`: Thanks for your comment. We will give the detailed statements about the unclear contents here.
>
> **(1) Firstly**, it needs to be known that we use different model architectures to solve different tasks. For Gaussian color image denoising, we use a similar framework with SwinIR. It is also the main framework as introduced in the main paper, which is shown in Fig.2(a). For real image denoising, we use the U-Net structure to design our network, which is similar to Restormer.
>
> **(2) Secondly**, we use a commonly-used training strategy when solving Gaussian color image denoising. The details can be found in **Section 4.1 of the main paper**.
>
> **(3) Thirdly**, we use the progressive training strategy when solving real image denoising to make fair comparisons with Restormer.  The details can be found in **Appendix C of the supplementary material**.

---

### Official Review · Reviewer_wRmT · 2022-10-23

**Confidence:** 5
**Correctness:** 4
**Technical Novelty And Significance:** 4
**Empirical Novelty And Significance:** 3
**Recommendation:** 8

**Clarity, Quality, Novelty And Reproducibility:**

The paper is organized well, stated clearly, and easy to follow. The work about attention retractable Transformer is novel and achieves promising results for several classic image restoration applications. The submitted code further makes it solid and reproducible.

**Strength And Weaknesses:**

Strengths

1. The paper is well-written and easy to follow. The idea about attention retractable Transformer is simple yet effective. The combination of sparse and dense attention help sense global and local information for high-quality image restoration.

2.The authors provide extensive experiments to show the effects of each proposed component and the whole method. They also discuss the differences between the proposed method and other related works, which highlight the novelty of the paper.

3.Some experimental curves (e.g., Figs. 4, 8, and 9) provide good visualization during the training. Those curve comparisons make the method more convincing.

4.The quantitative results are good. In Tabs. 2, 4, and 5, the proposed ART achieves the best PSNR/SSIM values in most cases. Such improvements over the previous methods are notable, especially for image SR. The authors also provide model size and flops comparisons, trying to keep fair comparisons with similar parameters and flops.

5.The visual comparisons (e.g., Figs. 5 and 7) show impressive results of the proposed method, which obtains obviously better outputs than others. Such comparisons are also provided in the supplementary file.

6.The supplementary file provides many useful experiments, like more ablation study results, analyses and performance comparisons with related work, real image denoising, and other quantitative and visual comparisons. All those supplements further demonstrate the effectiveness of the proposed method.

7.The authors provide demo code on an anonymous website, where the pretrained models and testing scripts are provided to reproduce results in the main paper and supplementary file. The code availability makes the work more solid and convincing.

Weaknesses

1.Some details are not clear enough. For example, in Tab. 7 of the supplementary file, real image denoising part, did the authors use the same pretrained model to test on SIDD and DND data? Or use different pretrained models for each dataset? Also, did the authors still use the architecture shown in Fig. 2(a) for real image denoising? (As I know, Restormer uses a different structure.) There are some training strategies in Restormer, like progressive training. Did the authors also use such a strategy?

2.It would be much better if the authors can provide some discussions or/and experiments of the method applied for other image restoration applications, like deraining and deblurring.

3.It would be much better if the authors could provide more curve and/or visual comparisons. For example, how about the validation curves and visual results between the proposed method and others, like Restormer.

**Summary Of The Paper:**

The paper investigates an efficient Transformer based method for high-quality image restoration. Specifically, the proposed Attention Retractable Transformer (ART) incorporates sparse and dense attention to sense larger receptive field. The alternative manner of those two attention blocks help capture global and local information. The ablation study and main comparisons support the claimed components well with SOTA performance.

**Summary Of The Review:**

Overall, the paper is high-quality with a novel method (ART). The experiments are very extensive and support the contributions well. The submitted code and models make the method more solid.

---

> ### Author Response · Authors · 2022-11-14
> **Response to Reviewer wRmT (denoted as R3)**
>
> `Q1`:Some details are not clear enough. For example, in Tab. 7 of the supplementary file, real image denoising part, did the authors use the same pretrained model to test on SIDD and DND data? Or use different pretrained models for each dataset? Also, did the authors still use the architecture shown in Fig. 2(a) for real image denoising? (As I know, Restormer uses a different structure.) There are some training strategies in Restormer, like progressive training. Did the authors also use such a strategy?
>
> `A1`: Thanks for your comment. We will give detailed statements about the unclear contents here. Besides, we will add more details about these parts in our paper.
>
> **(1) Firstly**, we use the same pre-trained model to test on SIDD and DND datasets. In detail, we only use 320 high-resolution images of SIDD to train our model. We do not use additional training data. Besides, we directly use the trained model to perform evaluations on SIDD and DND testing data.
>
> **(2) Secondly**, we do not use the architecture shown in Fig.2(a) to solve real image denoising. We choose to utilize the U-Net structure to design our proposed ART. The details can be found in **Appendix C of the supplementary material**.
>
> **(3) Thirdly**, we also use the progressive training strategy when solving real image denoising to make fair comparisons with Restormer.
>
>
> `Q2`:It would be much better if the authors can provide some discussions or/and experiments of the method applied for other image restoration applications, like deraining and deblurring.
>
> `A2`:  Thanks for this comment. It is meaningful and attractive to evaluate our proposed method on other image restoration tasks. Due to the time limitation, we have not conducted more experiments in our current work. During the rebuttal, we further evaluate the performance of our ART on a new task, which is Gaussian grayscale image denoising. We introduce the corresponding details and results here.
>
> **(1) Additional experiments.** We conduct additional experiments on Gaussian grayscale image denoising to compare our ART to Restormer. In order to make fair comparisons, we also use U-net structure to design our ART and train our models on the codebase of Restormer. We keep the same FFN expansion ratio and layers number of U-net network with Restormer. More details can be found **in Section D of our revised supplementary material.** We provide the results of PSNR-Params-FLOPs comparisons here. We calculate FLOPs assuming that the input size is 1×128×128. We also provide the results of SwinIR.
> | Method | Params (M) |FLOPs (G)| Set12 (15/25/50) | BSD68 (15/25/50) | Urban100 (15/25/50) |
> |-|-|-|-|-|-|
> | SwinIR|11.50|201.1| 33.36/31.01/27.91 | 31.97/29.50/26.58 | 33.70/31.30/27.98 |
> | Restormer|26.11|38.7| 33.42/31.08/28.00 | 31.96/29.52/**26.62** | 33.79/31.46/28.29 |
> | ART (ours)|20.82|43.8| **33.42/31.10/28.02** | **31.97/29.53**/26.60 | **33.89/31.68/28.56** |
>
> As we can see, our ART performs better than SwinIR and Restormer, especially on Urban100. It is seen that **our ART has comparable FLOPs with Restormer but owns 1.25× fewer model parameters**. The provided experimental results further demonstrate that our ART achieves better performance than Restormer. We also provide additional visual comparisons to validate the superiority of ART in **Figure 15-17 of the revised supplementary material part.**
>
>
> **(2) Further discussion.** Besides, in the conclusion part of our revised paper,  we will **discuss the future work in solving more low-level vision problems**, e.g., image deblurring, deraining, dehazing, enhancement, and so on. You can find the corresponding changed contents in our new version paper.
>
> `Q3`: It would be much better if the authors could provide more curve and/or visual comparisons. For example, how about the validation curves and visual results between the proposed method and others, like Restormer.
>
> `A3`: Thanks for your suggestions. We will **add the corresponding curve and visual comparisons in our revised paper**.
>
> **(1) Firstly**, we provide the validation curve comparisons of our ART and SwinIR under the ×2 SR training task. The total training iterations are 500k and the validating dataset is Set5. The detailed comparisons can be found in **Figure 10 of the revised supplementary material part**.
>
> **(2) Secondly**, we also **provide more visual comparisons with Restormer**. The corresponding results are provided as the new additional contents in the revised supplementary material part. You can find them in **Figure 15-17 of the revised supplementary material part**.

---

### Official Review · Reviewer_eLG6 · 2022-10-24

**Confidence:** 5
**Correctness:** 3
**Technical Novelty And Significance:** 3
**Empirical Novelty And Significance:** 3
**Recommendation:** 8

**Clarity, Quality, Novelty And Reproducibility:**

The paper is clearly organized, like some illustration figures, difference summary table. The motivation and method details are also clear. The method is novel and reproducible with the submitted code, models, and scripts.

**Strength And Weaknesses:**

The idea about using sparse and dense self-attention in Transformer is simple but effective. It is good to see such an investigation in image restoration and good performance for different tasks.

Also, the alternating application of sparse and dense attention blocks help capture the local and global information. The corresponding ablation study supports the claim well.

The main comparisons with recent methods for different tasks are extensive. The ART achieves quantitative gains and also show obviously better visual results in some challenging cases.

The supplementary file provides more experiments to show the effectiveness of the ART. They also submit the demo code and pretrained models, which make their work reproducible and more solid.

The paper is prepared pretty well and organized clearly. It is also easy to read and follow with the provided code.

Weaknesses:

The proposed method is claimed for image restoration. The authors show experiments on several image restoration tasks, like image super-resolution, color, and real image denoising, and JPEG compression artifact reduction. How would be the method performed for other restoration tasks? Like image deblurring, deraining, dehazing, and so on. It would be good if the authors can provide some additional results for one/some of those tasks. Or at least, the authors can briefly discuss them in the conclusion part.

In image denoising, the main architecture of ART is different from that of Restormer. Also, Restormer uses a progressive training strategy, which is not used in this paper. Do the authors think that ART would further benefit from the U-Net structure and progressive training used in Restormer? If so, why not use a similar structure and progressive training?


**Summary Of The Paper:**

The authors propose an Attention Retractable Transformer (ART), which incorporates sparse and dense attention to widen the receptive field size. The ART method achieves pretty good results in several image restoration tasks, like super-resolution, denoising, compression artifact reduction.

**Summary Of The Review:**

The authors propose a new Transformer-based image restoration method with several new components, which are demonstrated by the extensive ablation study and main experiments. Furthermore, the paper is well prepared and reproducible based on the submitted code and models.

---

> ### Author Response · Authors · 2022-11-14
> **Response to Reviewer eLG6 (denoted as R2)**
>
> `Q1`: The proposed method is claimed for image restoration. The authors show experiments on several image restoration tasks, like image super-resolution, color, and real image denoising, and JPEG compression artifact reduction. How would be the method performed for other restoration tasks? Like image deblurring, deraining, dehazing, and so on. It would be good if the authors can provide some additional results for one/some of those tasks. Or at least, the authors can briefly discuss them in the conclusion part.
>
> `A1`: Thanks for this comment. It is meaningful and attractive to evaluate our proposed method on other image restoration tasks. Due to the time limitation, we have not conducted more experiments in our current work. During the rebuttal, we further evaluate the performance of our ART on a new task, which is Gaussian grayscale image denoising. We introduce the corresponding details and results here.
>
> **(1) Additional experiments.** We conduct additional experiments on Gaussian grayscale image denoising to compare our ART to Restormer. In order to make fair comparisons, we also use U-net structure to design our ART and train our models on the codebase of Restormer. We keep the same FFN expansion ratio and layers number of U-net network with Restormer. More details can be found **in Section D of our revised supplementary material.** We provide the results of PSNR-Params-FLOPs comparisons here. We calculate FLOPs assuming that the input size is 1×128×128. We also provide the results of SwinIR.
> | Method | Params (M) |FLOPs (G)| Set12 (15/25/50) | BSD68 (15/25/50) | Urban100 (15/25/50) |
> |-|-|-|-|-|-|
> | SwinIR|11.50|201.1| 33.36/31.01/27.91 | 31.97/29.50/26.58 | 33.70/31.30/27.98 |
> | Restormer|26.11|38.7| 33.42/31.08/28.00 | 31.96/29.52/**26.62** | 33.79/31.46/28.29 |
> | ART (ours)|20.82|43.8| **33.42/31.10/28.02** | **31.97/29.53**/26.60 | **33.89/31.68/28.56** |
>
> As we can see, our ART performs better than SwinIR and Restormer, especially on Urban100. It is seen that **our ART has comparable FLOPs with Restormer but owns 1.25× fewer model parameters**. The provided experimental results further demonstrate that our ART achieves better performance than Restormer. We also provide additional visual comparisons to validate the superiority of ART in **Figure 15-17 of the revised supplementary material part.**
>
>
> **(2) Further discussion.** Besides, in the conclusion part of our revised paper,  we will **discuss the future work in solving more low-level vision problems**, e.g., image deblurring, deraining, dehazing, enhancement, and so on. You can find the corresponding changed contents in our new version paper.
>
>
> `Q2`: In image denoising, the main architecture of ART is different from that of Restormer. Also, Restormer uses a progressive training strategy, which is not used in this paper. Do the authors think that ART would further benefit from the U-Net structure and progressive training used in Restormer? If so, why not use a similar structure and progressive training?
>
> `A2`: Thanks for your proposed questions. We provide the reply as follows.
>
> **(1) Firstly**, we agree that our ART will further benefit from the U-Net structure and progressive training. To the best of our knowledge, the U-Net structure has been frequently used in recently proposed image restoration methods. The related works involve both CNN-based and Transformer-based methods, e.g., DRUNet [ref4], Uformer [ref5], and so on. The U-shaped structure can enable the network to capture multi-scale deep features while maintaining computational efficiency. For the progressive training, it is validated to bring performance improvement.
>
> **(2) Secondly**, we have designed a similar structure to compare our ART and Restormer on the Real Image Denoising task. The detailed experiments are presented in the appendix part, refer to **Appendix C**. We introduce the detailed experimental settings and provide quantitative comparisons with recent methods.
>
> **(3) Thirdly**, we will try to validate the experimental results of our ART on more image restoration tasks by using U-Net structure and progressive training. We will do this in our future work.
>
> [ref4] Kai Zhang, Yawei Li, Wangmeng Zuo, Lei Zhang, Luc Van Gool, and Radu Timofte. Plug-and-play image
> restoration with deep denoiser prior. TPAMI, 2021.
>
> [ref5] Zhendong Wang, Xiaodong Cun, Jianmin Bao, Wengang Zhou, Jianzhuang Liu, and Houqiang Li. Uformer: A
> general u-shaped transformer for image restoration. In CVPR, 2022.

---

### Official Review · Reviewer_dTRi · 2022-10-24

**Confidence:** 4
**Correctness:** 3
**Technical Novelty And Significance:** 3
**Empirical Novelty And Significance:** 2
**Recommendation:** 6

**Clarity, Quality, Novelty And Reproducibility:**

The paper is well organized and clearly presented. The non-local attention modules have been proposed and used in several works. This paper leverage sparse attention in the low-level vision task. Therefore, the novelty of paper is not significant.

**Strength And Weaknesses:**

Strength:
+  This paper introduces sparse attention mechanism to low-level vision task and enable tokens from sparse areas of the image to interact with each other to enlarge the receptive field of transformer.
+ The paper is well organized and clearly presented. In the paper, the differences and relations with other related works are discussed and analyzed.

Weakness:
- The improvement in performance is limited. In the SR results. The ART-s has similar parameter size and FLOPS as SwinIR.  The gain obtained by ART-s is insignificant in terms of PSNR and SSIM on dataset Set5, Set14, and B100. ART and ART+ outperforms the SwarIR by 0.1dB, but they are equipped with more parameters. For the image denosing task, the gap between the SwinIR and restormer is not remarkable.
- The proposed scheme introduces the features from dilated position to enable distant feature access, which benefits restoration of the images with repeated pattern, such as buildings in Urban100.  Table 2 and figure 4, the Urban100 are chosen to be compared with peer works.  The paper should provide more comprehensive result comparison to verify the advantage of ART.
- The description of the method could be further refined. E.g. The symbol “I” represents original input and intervals in the proposed method.


**Summary Of The Paper:**

This paper proposes Attention Retractable Transformer (ART) for image restoration, which integrates both dense and sparse attention modules in the transformer-based network. The dense attention block and sparse attention blocks emerge alternately to enable interactions among tokens extracted from a sparse area during restoration. The paper claims that the proposed ART enhances representation ability of Transformer while providing retractable attention on the input image. Extensive experiments are conducted on image super-resolution, denoising, and JPEG compression artifact reduction tasks.

**Summary Of The Review:**

The paper proposed sparse attention module and design interlaced attention blocks to leverage near and distant feature attention. The designed ART could achieve remarkable improvements on the images with repeat patterns, but it cannot prove its advantages on other general datasets.  The non-local attention modules have been proposed by other recognition works.

---

> ### Author Response · Authors · 2022-11-14
> **Response to Reviewer dTRi (denoted as R1) part 1**
>
> `Q1`: The improvement in performance is limited. In the SR results. The ART-s has similar parameter size and FLOPS as SwinIR. The gain obtained by ART-s is insignificant in terms of PSNR and SSIM on dataset Set5, Set14, and B100. ART and ART+ outperforms the SwarIR by 0.1dB, but they are equipped with more parameters. For the image denoising task, the gap between the SwinIR and restormer is not remarkable.
>
> `A1`: Thanks for your comments. We consider these contents carefully and give our replies as follows.
>
> **(1) About the performance in SR.** We provide the detailed SR results of our ART-S and ART in Tab.2 of the main paper. Experimental results show that our ART-S yields obvious performance gain over SwinIR on Urban100 and Manga109. On other three datasets, ART-S has comparable performance. It confirms that the quality of input images has an influence on model performance. We give some explanations here.
>
> **From the respect of model**, our method mainly brings performance improvement through enlarging receptive fields in Transformer. Therefore, it has the natural advantage to deal with large-size and richly textured images.
>
> **From the respect of data**, Set5, Set14, and B100 mainly include small-size or low-quality images (e.g., figures in B100 are compressed), thereby introducing difficulties into image restoration. The evaluation on these datasets has limited ability to mine the strengths of our model.  On the contrary, Urban100 and Manga109 own high-quality images. Our proposed ART-S can achieve better performance than SwinIR on these two benchmarks. It reveals that our method has superiority over SwinIR. As the scale size grows, our ART-S still outperforms SwinIR, which further validates the advantages of our method.
>
> **In conclusion, our proposed model can still achieve remarkable performance with fewer parameters.** Our ART-S achieves obvious performance gain over SwinIR on those datasets with high-quality images.
>
> **(2) About the performance in image denoising.** We report the evaluation results on gaussian color image denoising in Tab.4 of the main paper. Some important statements and comparisons are provided here.
>
> **The Kodak24 results of Restormer are different from those generated by their officially provided pre-trained models.** It can be found that Restormer only outperforms our method on Kodak24 datasets. In fact, the results of Restormer on Kodak24 are not consistent with the true testing results using their officially provided pre-trained models and code. Besides, we follow the complete training settings in the codebase of Restormer and train a new Restormer model on Gaussian color image denoising with 15 noise level. We also get lower testing results on Kodak24 benchmark. Therefore, we report the new results on Kodak24 here, which are obtained by testing with their officially provided models and code. We also report results on other datasets, which are also from the testing with their pre-trained models. We also provide the officially reported results of SwinIR.
> | PSNR results (dB)| CBSD68 (15/25/50) | Kodak24 (15/25/50) | McMaster (15/25/50) | Urban100 (15/25/50) |
> |-|-|-|-|-|
> | SwinIR | 34.42/31.78/28.56| 35.34/32.89/29.79| 35.61/33.20/30.22| 35.13/32.90/29.82|
> | Restormer| 34.40/31.79/28.60| 35.35/32.93/29.87| 35.61/33.34/30.30| 35.12/32.94/30.01|
> | ART (ours)| **34.46/31.84/28.63** | **35.39/32.95/29.87** | **35.68/33.41/30.31** | **35.29/33.14/30.19** |
>
> As we can see, **our ART achieves the best performance on all benchmark datasets across three noise levels.** Besides, we claim that all the testing results of ART can be reproduced by using our provided pre-trained models and code. Therefore, our proposed method is validated to be superior over SwinIR and Restormer on image denoising.

---

> ### Author Response · Authors · 2022-11-14
> **Response to Reviewer dTRi (denoted as R1) part 2**
>
> `Q2`: The proposed scheme introduces the features from dilated position to enable distant feature access, which benefits restoration of the images with repeated pattern, such as buildings in Urban100. Table 2 and figure 4, the Urban100 are chosen to be compared with peer works. The paper should provide more comprehensive result comparison to verify the advantage of ART.
>
> `A2`: Thanks for your comments. We demonstrate that our ART can also perform well on other data with few repeated patterns, e.g., Manga109. According to your questions, we give our replies as follows.
>
> **(1) About the results in Table 2.** We guess you may refer to the results in Table 3, because Table 2 has included testing results on five benchmark datasets. The contents of Table 3 are about the model size comparisons with recent leading methods. Evaluation results are based on ×4 SR tasks. Therefore, we provide more detailed results in Table 3 here. We choose to compare parts of the involved methods in Table 3. PSNR scores are reported and FLOPs are calculated assuming the output size is 3×640×640.
> | Method | Params (M) |FLOPs (G)| Set5| Set14 |B100|Urban100|Manga109|
> |-|-|-|-|-|-|-|-|
> |EDSR|43.09 | 1,286|32.46 | 28.80 | 27.71| 31.02|26.64 |
> |CSNLN| 7.16 | 103,640 |32.68 | 28.95 | 27.80 | 31.43 |27.22|
> | SwinIR|11.90|336| 32.92 | 29.09 | 27.92 | 32.03|27.45|
> | ART-S|11.87|392| 32.86 | 29.09 | 27.91| 32.13|27.54 |
> | ART|16.55|782| 33.04 | 29.16 | 27.97 | 32.31|27.77|
>
> As we can see, our proposed methods also achieve promising performance on the datasets with few repeated patterns, e.g., Manga109.
>
> **(2) About the results in Figure 4.** We put three sub-figures in Figure 4 in which evaluation results are based on Urban100. We could provide corresponding figures using testing results on Manga109. Note that Manga109 includes images with few repeated patterns. In our original paper, we have provided the PSNR comparisons on Manga109 among SwinIR, ART-S, and ART in **Figure 8 of Appendix part.** Therefore, we will update two other new figures in our revised supplementary material. You can find the new contents in **Figures 8 and 9 of the revised supplementary material.**
>
> **(3) More comprehensive results.** We provide more testing results on other extended datasets here.
>
> **Firstly,** we evaluate our ART on extra datasets Manga109 for Gaussian color image denoising task. In order to make fair comparisons, we download the officially provided pre-trained models of SwinIR and Restormer and use them under the same testing settings as ART. We report the PSNR and SSIM scores here.
> | Testing on Manga109 | noise level | 15 | 25 | 50 |
> |-|-|-|-|-|
> |SwinIR|PSNR (dB)/SSIM| 36.19/0.9494 | 34.25/0.9319 | 31.52/0.9093 |
> |Restormer|PSNR (dB)/SSIM| 36.27/0.9506 | 34.26/0.9322 | 31.58/0.9045 |
> |ART (ours)|PSNR (dB)/SSIM|**36.51/0.9521** | **34.46/0.9340** | **31.73/0.9058** |
>
> As we can see, our proposed ART achieves obvious performance gains across all noise levels. These results further verify the advantage of ART. Besides, it reveals that our method can also perform well in restoring images with few repeated patterns.
>
> **Secondly,** we evaluate our ART on extra datasets Urban100 and Manga109 for JPEG compression artifact reduction task. To make fair comparisons, we also use the officially provided pre-trained models of SwinIR to obtain results with the same testing settings. We report the PSNR and SSIM scores here.
> | Dataset | q | Metrics | SwinIR | ART (ours) |
> |-|-|-|-|-|
> |Urban100|10|PSNR (dB)/SSIM| 30.55/0.8840 | **30.85/0.8886** |
> |Urban100|30|PSNR (dB)/SSIM| 34.57/0.9417 | **34.79/0.9432** |
> |Urban100|40|PSNR (dB)/SSIM| 35.50/0.9508 | **35.71/0.9520** |
> |Manga109|10|PSNR (dB)/SSIM| 34.04/0.9333 | **34.15/0.9343** |
> |Manga109|30|PSNR (dB)/SSIM| 38.17/0.9638 | **38.23/0.9641** |
> |Manga109|40|PSNR (dB)/SSIM| 39.09/0.9694 | **39.13/0.9696** |
>
> As we can see, our ART achieves better performance than SwinIR across all compression qualities on both Urban100 and Manga109 datasets. The experimental results further verify the advantages of ART.
>
> **Thirdly,** we add the validation curve comparisons of ART and SwinIR on SR ×2 task. The validating dataset is Set5. You can find that our ART has better performance in **Figure 10 of the revised supplementary material.**

---

> ### Author Response · Authors · 2022-11-14
> **Response to Reviewer dTRi (denoted as R1) part 3**
>
> `Q3`: The description of the method could be further refined. E.g. The symbol “I” represents original input and intervals in the proposed method.
>
> `A3`: Thanks for correcting this detail. **We will change the symbol of the original input to avoid misunderstandings.** You can find the changed contents in our revised paper. Besides, we will check our paper again carefully.
>
>
>
> `Q4`: The non-local attention modules have been proposed and used in several (recognition) works. This paper leverage sparse attention in the low-level vision task. Therefore, the novelty of paper is not significant.
>
> `A4`: Thanks for your comment. We demonstrate that our proposed method based on sparse attention has differences from recent high-level vision methods. We will give our replies as follows.
>
> **(1) Analyses and discussion.** We introduce the differences of our method in Section 1 and provide a detailed discussion in Section 3.3 of the main paper. We also provide more analyses in **Appendix B of the supplementary material**. We give the analyses and discussion here.
>
> **Firstly,** we design our ART to solve low-level vision problems. Different from the solutions of high-level tasks, our method focuses on learning more pixel-level information rather than semantic-level information.
>
> **Secondly,** the designs of sparse attention blocks are different. For example, GG-Transformer [ref1] uses the adaptively-dilated partitions to calculate self-attention and CrossFormer [ref2] utilizes the cross-scale long-distance attention mechanism. As the layers get deeper, the interval of tokens from the sparse area becomes smaller and the channels of tokens become deeper. It is hard for these tokens to learn pixel-level information. In contrast, the tokens in our ART own unchanged intervals and channels and thus bring fine-grained global representations.
>
> **Thirdly,** we design our ART with different model architectures. Existing high-level vision methods mainly build the pyramid model structure. Our ART enjoys an isotropic structure and thus retains more patch-level information, which is important for recovering clean images. Besides, we design the long-distance residual connection, which enables features of deep layers to reserve more low-frequency information from shallow layers.
>
> **(2) Experimental verification.** In **Appendix B.2 of the supplementary material,** we provide the comparative experiments to demonstrate that existing model designs of high-level vision methods are not suitable for low-level vision tasks. We consider CrossFormer as a representative work and make comparisons with it. We also give the details here.
>
> **For experimental settings,** we keep the corresponding parameters in ART and CrossFormer the same. We use the same training data and settings. We train both models for 300k iterations. More details can be found in the paper.
>
> **For experimental results,** we provide quantitative comparisons in Table 6 and curve comparisons in Figure 9 of the Appendix part. You can find them in the supplementary material. We also report the PSNR (dB) results here.
> | Method | scale | Set5 | Set14 | B100 | Urban100 | Manga109 |
> |-|-|-|-|-|-|-|
> | CrossFormer| ×2 | 38.20 | 34.04 | 32.32 | 32.91 | 39.28 |
> | ART (ours)| ×2 |**38.39**|**34.33**|**32.49**|**33.70**|**39.88**|
>
> As we can see,  our method achieves obvious performance gains over CrossFormer. **With few modifications, existing high-level model designs with sparse attention are not suitable for solving image restoration problems.**
>
> **(3) Conclusion.** In our paper, we propose the sparse attention to compensate the defect of mainly using dense attention in existing Transformer-based image restoration networks, e.g., SwinIR [ref3], and so on. We provide extensive experiments to verify the advantage of our method. Besides, **analyses and experimental verifications can support that our method is different from recent high-level vision works.** Therefore, we think the novelty of our proposed method is significant.
>
> [ref1] Qihang Yu, Yingda Xia, Yutong Bai, Yongyi Lu, Alan L Yuille, and Wei Shen. Glance-and-gaze vision transformer. In NeurIPS, 2021.
>
> [ref2] Wenxiao Wang, Lu Yao, Long Chen, Binbin Lin, Deng Cai, Xiaofei He, and Wei Liu. Crossformer: A versatile vision transformer hinging on cross-scale attention. In ICLR, 2022.
>
> [ref3] Jingyun Liang, Jiezhang Cao, Guolei Sun, Kai Zhang, Luc Van Gool, and Radu Timofte. Swinir: Image restoration using swin transformer. In ICCVW, 2021.

---

### Author Response · Authors · 2022-11-14
**Response to all reviewers and area chairs for a brief summary**

We thank all reviewers for their precious review time and valuable comments. We would like to give a brief summary here.

**1. What we have done during the rebuttal phase.**

(1) We give detailed replies to all the concerns given by Reviewer dTRi. We analyze the advantages of our proposed method and provide **additional experimental results to verify its effectiveness and robustness.** We present more comprehensive results on other general datasets, which further demonstrate that our method can **achieve significant performance improvement over existing SOTA methods**. Besides, we provide abundant statements about the novelty of our method. **Analyses and experimental verifications can support that our paper is innovative.**

(2) We reply to the comments about using our method to solve more image restoration tasks. **We conduct new experiments about solving Gaussian grayscale image denoising task**. We provide quantitative, visual, and model complexity comparisons to show the superiority of our method. Note that we also provide the corresponding models and code in the anonymous website. We promise to further extend our work in the future.

(3) **We explain some unclear parts** in our paper and provide more details. We give the details about the model and training settings when using our method to solve real image denoising task.

(4) **We take some valuable suggestions** proposed by reviewers, e.g., correcting the symbols, adjusting contents in the main paper, providing open-source training details, and so on.

**2. The modifications of the main paper and supplementary material.**

(1) **Main Paper.** We consider the questions and comments of all reviewers and modify our main paper. The main modifications are three points. **Firstly,** we correct the symbols of some descriptors to avoid misunderstanding. **Secondly,** we discuss more related works. **Thirdly,** we discuss future work about solving more image restoration tasks in the conclusion part. **Note that we revise the text contents and mark them in orange.** You can easily find them.

(2) **Supplementary Material.** **Firstly,** we provide more evaluation results on Manga109 in additional ablation studies. **Secondly,** we conduct more experiments on a new task: Gaussian grayscale image denoising. **Thirdly,** we provide more validation curve and visual comparisons.

---

### Decision · Program_Chairs · 2023-01-20

**Decision:**

Accept: notable-top-25%

**Justification For Why Not Higher Score:**

While the proposed approach is interesting and contributes to improved image restoration with transformers, the importance for the machine learing community could be stronger if restoration beyond images were addressed.

**Justification For Why Not Lower Score:**

The paper makes a strong technical contribution and is well evaluated. It deserves to be presented as a spotlight!

**Metareview: Summary, Strengths And Weaknesses:**

This paper presents a variant of transformers, an attention retractable Transformer (ART) that allows for accurate image restoration. The proposed model is innovative. It combines dense and sparse attention modules. The proposed sparse attention can allow token interactions in sparse image regions and thus enlarge the receptive field of the module. The combination of sparse and dense modules allows for global and local interactions while being tractable.

The proposed model is validated on substancial experiments for several image restoration settings. Even more results are provided in the supplementary material. Yet, results on other restoration tasks (beyond images) ar not provided. Source code is also given in the supp. mat.. .

All reviewers give positve scores and agree that the paper not only proposes an interesting module for image restoration but is also well-written, motivated and illustrated.
Yet, the reviewers also find that some details should be more clearly described. AC recommends to thoroughly go through the remaining questions and provide clarifications in the supplementary material.


**Note From Pc:**

if the above contains the word "oral" or "spotlight" please see: "oral" presentation means -> notable-top-5% and "spotlight" means -> notable-top-25%. As stated in our emails, we are disassociating presentation type from AC recommendations